# Oracle Guided Image Synthesis with Relative Queries

**Alec Helbling, Christopher John Rozell, Matthew O'Shaughnessy & Kion Fallah**
School of Electrical and Computer Engineering
Georgia Institute of Technology
`{alechelbling, crozell, moshaughnessy6, kion}@gatech.edu`

## Abstract

Isolating and controlling specific features in the outputs of generative models in a user-friendly way is a difficult and open-ended problem. We develop techniques that allow a user to generate an image they are envisioning in their head by answering a sequence of relative queries of the form *"do you prefer image $a$ or image $b$?"* Our framework consists of a Conditional VAE that uses the collected relative queries to partition the latent space into preference-relevant features and non-preference-relevant features. We then use the user's responses to relative queries to determine the preference-relevant features that correspond to their envisioned output image. Additionally, we develop techniques for modeling the uncertainty in images' predicted preference-relevant features, allowing our framework to generalize to scenarios in which the relative query training set contains noise. [1]

## 1 Introduction

Deep generative models have recently demonstrated the ability to transform samples from a latent distribution to high-fidelity, photorealistic images (Oussidi & Elhassouny, 2018). However, the latent attributes learned by these models do not necessarily correspond to intuitive features that a user will implicitly understand. This makes rendering images with specific semantic features difficult for end-users. In this paper, we outline a framework for using relative queries to guide the generative process of Variational Autoencoders (VAEs) (Kingma & Welling, 2014) to allow end-users to render images that they can envision but not directly describe.

Many image features, called relative features, are best understood by comparing pairs of images (Parikh & Grauman, 2011). For example, most people would likely find it difficult to quantify how *angry* a person appears, but are easily able to articulate which of two people appears *angrier*. Our core contribution is a framework that allows a user to generate images with specific relative features by simply answering a sequence of relative queries of the form *"do you prefer image $a$ or image $b$?"* We learn a Conditional VAE model (Kingma et al., 2014) which can conditionally generate

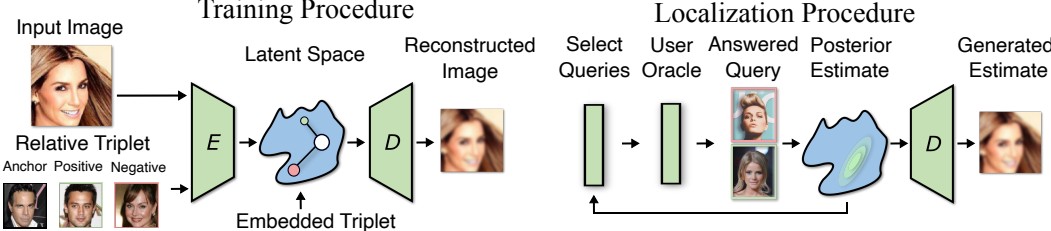

**Figure 1:** *Left:* The VAE is trained with a combination of unsupervised image data and weakly supervised triplet data to encode relative attributes. *Right:* The paired comparison localization procedure is used to estimate a posterior distribution in latent space representing the user's preference. The posterior mean is then decoded to an estimate image.

---

[1]Code available at `https://github.com/helblazer811/oracle-guided-image-synthesis`

images from a set of preference-relevant "relative" features $\mathbf{r} \in \mathbb{R}^z$ and latent reconstructive features $\mathbf{z} \in \mathbb{R}^z$. The VAE encoder $E \colon \mathbf{x} \to (\mathbf{r}, \mathbf{z})$ learns to infer both relative features and reconstructive features given an image, and the decoder $D \colon (\mathbf{r}, \mathbf{z}) \to \mathbf{x}$ learns to conditionally generate images from $\mathbf{r}$ and $\mathbf{z}$. We then use a sequence of relative queries to deduce a distribution of user preference over these relative features (Canal et al., 2019; Jamieson & Nowak, 2011), and sample from this distribution to generate images that match user preferences (See Figure 1).

Using relative queries unlocks the ability to model intuitive concepts that are difficult to explicitly describe or quantify (Thurstone, 1927; David, 1963). Prior work has used these types relative comparisons to perform tasks such as learning an embedding (Agarwal et al., 2007; Tamuz et al., 2011) or searching an embedding of relative attributes (Canal et al., 2019; Jamieson & Nowak, 2011; Davenport, 2013); we build upon this literature by using relative queries to control the generative process of a VAE. We train our VAE to map images to a set of relative attributes using the triplet loss (Karaletsos et al., 2015). Additionally, we train our model to be robust to noisy relative queries by quantifying the uncertainty in the predicted relative features (Warburg et al.). We demonstrate the success of our approach with experiments on the Morpho-MNIST dataset (Castro et al., 2019).

---

**Algorithm 1** Relative Comparison Search

**Input:** Decoder $D(\mathbf{r}, \mathbf{z})$, Oracle $O(\mathbf{q})$,
Response Model $R$, Query Selector $S$,
Number of Queries $T$
Initialize query set $Q = \{\}$
**for** $i = 1$ **to** $T$ **do**
    Generate query from $\mathbf{q}_i = \{\mathbf{p}, \mathbf{n}\}$ from $S$
    Answer $\mathbf{q}_i$ with the oracle $\mathbf{o}_i = O(\mathbf{q}_i)$
    Add the *answered query* to $Q = Q \cup \mathbf{o}_i$
    Find $p(\mathbf{r}_* \mid Q)$ with response model $R$
**end for**
Sample an arbitrary vector $\widehat{\mathbf{z}} \sim \mathcal{N}(0, I_z)$
Decode the posterior mean $\widehat{\mathbf{x}}^* = D(\mathbf{r}_{*\mu}, \widehat{\mathbf{z}})$
**Output:** decoded image $\widehat{\mathbf{x}}^*$

Final Search Estimates

Figure 2: Over eight trials we show 1) the decoded final estimate vector, 2) the Nearest Neighbor image with relative features closest to our final estimate vector, and 3) the ground truth image. We use the Bayesian VAE model with noise using the triplet response model. Both the decoded and nearest neighbor images seem similar to the ground truth.

## 2 DEDUCING USER PREFERENCES FROM RELATIVE QUERIES

We first describe our method for determining the relative features $\mathbf{r}_*$ corresponding to a user's envisioned image $\mathbf{x}^*$. We search the space of relative features $\mathbf{r}$ by asking a sequence of queries

$$Q = \{(\mathbf{x}_p^i, \mathbf{x}_n^i) \mid \mathbf{x}_p^i \prec \mathbf{x}_n^i, \text{ for } N \text{ queries}\}, \tag{1}$$

where $\mathbf{x}_p^i \prec \mathbf{x}_n^i$ indicates that in the $i^{\text{th}}$ query the user prefers $\mathbf{x}_p^i$ to $\mathbf{x}_n^i$. We assume that $\mathbf{x}_p^i \prec \mathbf{x}_n^i$ implies that $|\mathbf{r}_p^i - \mathbf{r}_*| < |\mathbf{r}_n^i - \mathbf{r}_*|$ in relative attribute space. Following Canal et al. (2019), we model the probability that a user will select an image with relative features $\mathbf{r}_p^i$ over an image with features $\mathbf{r}_n^i$ using the logistic response model

$$p(Q_i \mid \mathbf{r}_*) = \sigma(k(\mid \mathbf{r}_* - \mathbf{r}_n^i \mid^2 - \mid \mathbf{r}_* - \mathbf{r}_p^i \mid^2)), \tag{2}$$

where $\sigma(x)$ denotes the logistic function and $k$ is a tuneable noise constant corresponding to the confidence in a user responses. Using the information from the queries $Q$, we infer a probability distribution $p(\mathbf{r}_* \mid Q)$ over relative features. We model the relationship between a single query response and $\mathbf{r}_*$ through the posterior $p(Q_i \mid \mathbf{r}_*)$ where $Q_i = (\mathbf{x}_p^i, \mathbf{x}_n^i)$, and combine the information from multiple queries using a recursive application of Bayes rule. Using MCMC with a Gaussian prior $p(\mathbf{r}) \sim \mathcal{N}(0, I_r)$, we sample from $p(\mathbf{r}_*|Q)$ to predict $\mathbf{r}_*$. We concatenate $\mathbf{r}_*$ with an arbitrary vector $\mathbf{z} \sim \mathcal{N}(0, I_z)$ and decode this using our VAE decoder to generate an approximation of the user's envisioned image $\widehat{\mathbf{x}}^*$. This procedure is shown in Algorithm 1 and Figure 1.

## 3 Relative Attribute Conditional Variational Autoencoder

We now describe our procedure for training the Conditional VAE (Kingma et al., 2014) that allows us to model the conditional likelihood $p_\theta(\mathbf{x} \mid \mathbf{r}, \mathbf{z})$, which we can use to generate images $\mathbf{x}$ conditioned on relative features $\mathbf{r}$ and reconstructive features $\mathbf{z}$. We restrict the relative attributes $\mathbf{r}$ to be a contextually relevant set of features; in a face generation task, for example, this may include hair or eye color. The reconstructive features $\mathbf{z}$ encode properties necessary to generate images (such as lighting or pose), but which a user has no particular interest in controlling.

There is no guarantee that an unsupervised VAE will by default learn to encode relative features $\mathbf{r}$ so that points are ordered in a way that respects a user's perceptual similarity. To solve this problem we optimize a variation of the triplet loss (Schroff et al., 2015) (Figure 1), which allows us to learn an embedding of features $\mathbf{r}$ that satisfy triplet constraints of the form *"anchor $\mathbf{x}_a$ is more similar to a positive image $\mathbf{x}_p$ than a negative image $\mathbf{x}_n$."* A core benefit of using triplet comparisons is that they can be indirectly inferred from easily-available sources like search engine mouse click data (Joachims, 2002; Joachims et al., 2007), so they can be applied in more general contexts than explicit quantitative attributes. We add the triplet loss $L_t(\mathbf{r_a}, \mathbf{r_p}, \mathbf{r_n})$ to our objective function as

$$\mathbb{E}_{q_\phi(\mathbf{z},\mathbf{r}|\mathbf{x})}[\log p_\theta(\mathbf{x} \mid \mathbf{z}, \mathbf{r})] - D_{KL}(q_\phi(\mathbf{z}, \mathbf{r} \mid \mathbf{x}) \parallel p(\mathbf{z}, \mathbf{r})) - L_t(\mathbf{r_a}, \mathbf{r_p}, \mathbf{r_n}). \quad (3)$$

## 4 Relative comparison search with uncertainty quantification

We found that when data is limited or noisy it is very rarely feasible to perfectly learn an embedding that matches relative features $\mathbf{r}$. However, the logistic response model of Equation 2 assumes we know the exact features $\mathbf{r}$ for each image. If our VAE encoder incorrectly predicts features $\mathbf{r}$ for a pair of query images $(\mathbf{x}_p, \mathbf{x}_n)$ the logistic response model can be overconfident in the predicted locations of the user's preferred features $\mathbf{r}_*$, leading to the generation of images that do not satisfy the user's preferences. We address this problem by modeling the uncertainty in predicted relative features $\mathbf{r}$. For each image $\mathbf{x}$ we use our VAE encoder to predict a distribution $\mathcal{N}(\mathbf{r}_\mu, \mathbf{r}_\sigma)$, using the variance $\mathbf{r}_{\sigma^2}$ to encode the uncertainty of the exact relative features corresponding to an image. To perform this uncertainty quantification, we optimize the Bayesian Triplet Loss (BTL) (Warburg et al.). Using the BTL we represent each triplet item as a Gaussian over relative attributes $\mathbf{r}$. We predict the distribution for a triplet $\mathbf{t} = (\mathbf{r}_a, \mathbf{r}_p, \mathbf{r}_n)$ as

$$p(\mathbf{t} \text{ is satisfied}) = p(\boldsymbol{\tau} < 0) \approx p(|\mathbf{r}_a - \mathbf{r}_p|^2 - |\mathbf{r}_a - \mathbf{r}_n|^2 < 0). \quad (4)$$

Here $\boldsymbol{\tau}$ is a random variable representing $|\mathbf{r}_a - \mathbf{r}_p|^2 - |\mathbf{r}_a - \mathbf{r}_n|^2$ given the parameters of the distributions of the positive $\mathbf{r}_p = \mathcal{N}(\mathbf{p}_\mu, \mathbf{p}_\sigma)$, negative $\mathbf{r}_n = \mathcal{N}(\mathbf{n}_\mu, \mathbf{n}_\sigma)$, and the ideal point $\mathbf{r}_* = \mathcal{N}(\mathbf{r}_{*\mu}, \mathbf{r}_{*\sigma})$ (derivation in Appendix A). $p(\boldsymbol{\tau} < 0)$ corresponds to the probability that $\mathbf{r}_p$ is closer to $\mathbf{r}_*$ than $\mathbf{r}_n$. We optimize the negative log likelihood of $p(\boldsymbol{\tau} < -m)$

$$L_t(\mathbf{r_a}, \mathbf{r_p}, \mathbf{r_n}) = -\log(p(\boldsymbol{\tau} < -m)), \quad (5)$$

where $m$ corresponds to the triplet margin (Warburg et al.). We replace the logistic response model with a response model with uncertainty quantification, and model the posterior $p(Q_i \mid \mathbf{r}_*)$ as

$$p(\text{user chooses } \mathbf{r}_p \text{ over } \mathbf{r}_n \text{ given } \mathbf{r}_*) = p(\boldsymbol{\tau} < 0) \approx p(|\mathbf{r}_* - \mathbf{r}_p|^2 - |\mathbf{r}_* - \mathbf{r}_n|^2 < 0). \quad (6)$$

This model, which we call the Bayesian Triplet Response Model (BTRM), allows us to account for the uncertainty in the predicted relative attributes $\mathbf{r}$. This allows our technique for predicting $p(\mathbf{r}_*|Q)$ to be robust to imperfect predictions of an image's relative attributes $\mathbf{r}$.

## 5 Experiments

The focus of our experiments is to 1) demonstrate that we can generate images that align with user preferences, and 2) investigate the impact of noise in the triplet dataset on localization performance. We use the simple MorphoMNIST dataset, which consists of MNIST-like digits associated with metadata features (such as slant and thickness) that allow us to quantitatively evaluate the performance of our method (Castro et al., 2019). After using this metadata to generate training triplets, we withhold the exact quantitative values from the model and use it only for testing purposes. We

Table 1: Network architecture reconstruction performance and triplet loss performance

|  | Percentage of Triplets Satisfied | | Reconstruction Error | |
|---|---|---|---|---|
|  | Without Noise | With Noise | Without Noise | With Noise |
| Bayesian | 91.6 | 82.3 | 0.036 | 0.048 |
| Traditional | 89.8 | 78.7 | 0.037 | 0.050 |
| Unsupervised | 77.9 | 76.9 | 0.034 | 0.034 |

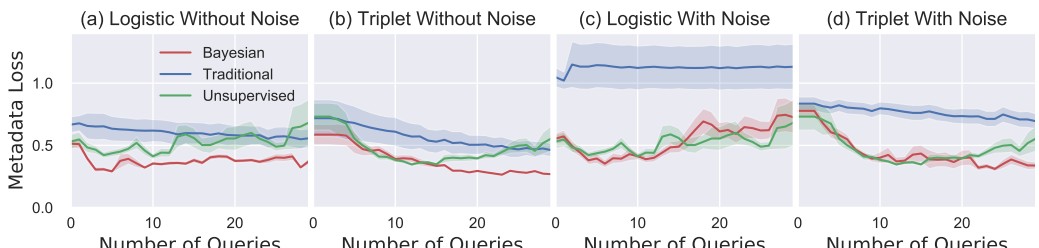

Figure 3: Shows the search performance of various VAE objective functions, response models, and noise conditions. The vertical axis represents the Metadata Loss, which measures the distance between the decoded mean in metadata space and the ground truth image's metadata. Each plot shows the localization performance for four different objective functions.

train each of our models using the objective of Equation 3; for simplicity, our experiments use only the digit 1, a six-dimensional relative attribute space, and a zero-dimensional reconstructive space. We compare the localization performance of three objective functions: the BTL ("Bayesian"), the traditional triplet loss ("Traditional"), and the unsupervised VAE objective with no triplet constraint ("Unsupervised"). We also investigate two response models during our localization task: the Logistic Response Model and the Bayesian Triplet Response Model. Finally, we investigate the impact of noise in the triplet collection process by adding Gaussian noise to the MorphoMNIST metadata, then generating triplet data from the noisy metadata. We follow Algorithm 1 using a synthetic oracle meant to simulate a user, and ask 30 queries for 20 trials. See Appendix B for experimental details.

Our quantitative results (Figure 3) show that the "Bayesian" model with the triplet response model outperforms other approaches at the localization task. As expected, the logistic response model fails in the presence of noise compared to the triplet response model. In the noiseless setting the triplet response model still outperforms the logistic model, likely due to better alignment between the triplet objective and triplet response model. We believe the unsupervised model gets closer to the ground truth for several queries and then diverges because the response model is overconfident in its estimate of $p(\mathbf{r}_* | Q_i)$, which causes convergence to incorrect local minima. It is interesting that the triplet response model works relatively well even for the Unsupervised and Traditional objective functions, despite those functions not attempting to quantify uncertainty. This is likely due, in part, to the encoded means for each image being accurate. There may also be a degree of unsupervised uncertainty quantification from the VAE objective. While we can achieve satisfactory qualitative localization performance (Figure 2), there is a slight trade-off in reconstructive performance for the supervised models, particularly in the presence of noise.

## 6 DISCUSSION

Our experiments provide a preliminary demonstration of our framework's efficacy in allowing users to guide the image synthesis process of VAEs using relative comparisons. There are several avenues for future work. First, when using higher-dimensional latent spaces we found that information from the relative subspace often "leaked" into the reconstructive subspace, causing the decoder to ignore the relative features in favor of the reconstructive features, and leading to poor generative performance. One potential solution to this problem may be to use cycle consistency (Jha et al., 2018), which may explicitly encourage independence between the reconstructive and relative features. Second, constraining the dimensionality of the relative features can reduce computational

cost; constraints that allow a low dimensional relative feature embedding to be learned dynamically (e.g., Veit et al. (2017)). Finally, active query selection techniques (e.g., Canal et al. (2019)) can reduce the number of queries required to converge on a quality image estimate by minimizing redundant queries.

## 7 ACKNOWLEDGMENTS

We would like to thank the generous support and guidance of the members of the Georgia Tech SIPLab. This work was partially supported by Georgia Tech's President's Undergraduate Research Award (PURA), the NSF CAREER award CCF-1350954, and the NSF Research Experiences for Undergraduates (REU) program.

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

## A CLOSED-FORM EXPRESSION FOR BAYESIAN TRIPLET LIKELIHOOD

Warburg et al. shows that the distribution $p(\boldsymbol{\tau} < -m)$, where $m$ corresponds to the triplet-margin, has a closed form representation with mean

$$\boldsymbol{\mu}_\tau = \boldsymbol{\mu}_p^2 + \boldsymbol{\sigma}_p^2 - \boldsymbol{\mu}_n^2 + \boldsymbol{\sigma}_n^2 - 2\boldsymbol{\mu}_*(\boldsymbol{\mu}_p - \boldsymbol{\mu}_n) \tag{7}$$

and variance

$$\begin{aligned}
\boldsymbol{\sigma}_\tau^2 = {}& 2(\boldsymbol{\sigma}_p^2(\boldsymbol{\sigma}_p^2 + 2\boldsymbol{\mu}_p^2) + \boldsymbol{\sigma}_n^2(\boldsymbol{\sigma}_n^2 + 2\boldsymbol{\mu}_n^2) - 4\boldsymbol{\sigma}_*^2\boldsymbol{\mu}_p\boldsymbol{\mu}_n \\
& + 2\boldsymbol{\mu}_*(\boldsymbol{\mu}_*(\boldsymbol{\mu}_p^2 + \boldsymbol{\mu}_n^2) - 2\boldsymbol{\mu}_p\boldsymbol{\sigma}_p^2 - 2\boldsymbol{\mu}_n\boldsymbol{\sigma}_n^2) \\
& + 2(\boldsymbol{\sigma}_*^2 + \boldsymbol{\mu}_*^2)((\boldsymbol{\sigma}_p^2 + \boldsymbol{\mu}_p^2) + (\boldsymbol{\sigma}_n^2 + \boldsymbol{\mu}_n^2))).
\end{aligned} \tag{8}$$

## B DETAILS OF EXPERIMENTAL SETUP

### B.1 SYNTHETIC ORACLE

We use a synthetic oracle $O(a, b)$ to evaluate our framework. For each query our oracle is given the ground truth image and metadata, as well as two choice images along with their metadata. The oracle selects as "preferred" the image whose metadata vector is closest in $\ell_2$ distance to the metadata vector of the ground truth image. This procedure simulates a person who answers queries based on a few high-level features such as digit slant, width, and height.

### B.2 LOCALIZATION PROCEDURE

For our experiments we ran 20 trials of 30 queries each for each technique. We use the following procedure for each localization trial:

1. Randomly select an image and its corresponding metadata $(\mathbf{x}^*, \mathbf{r}_*)$ from our test set, which we call the ground truth image

2. Follow Algorithm 1 using the synthetic oracle described above as $O$

   (a) During each iteration record current posterior mean estimate $\widehat{\mathbf{x}}^*$

   (b) Decode the posterior mean estimate to an image

   (c) Measure the MorphoMNIST metadata values of the given image and measure the distance to the ground truth metadata

### B.3 NEURAL NETWORK ARCHITECTURE

We used a simple VAE architecture with the following layers:

1. We had 4 units each with:

   (a) A Convolutional Layer

   (b) A Batchnorm Layer

   (c) A ReLU layer

2. 2 Linear layers each followed by a ReLU and BatchNorm

3. 6 Latent Units

4. 2 Linear Layers

5. 4 units each with:

   (a) A Transposed Convolution Layer

   (b) A Batchnorm Layer

   (c) A ReLU layer

## B.4   HYPERPARAMETER SETTINGS

The core hyperparemters of our system and values are as follows:

1. Learning Rate (0.0001)
2. Epochs (100)
3. Batch Size (256)
4. Optimizer Type (Adam)
5. Latent Dimension (6)
6. KL Divergence Beta (0.01)
7. Triplet Beta (0.1)
8. Triplet Margin (0.1)
9. Adam Betas (0.9, 0.999)

