# OpenReview forum: "Oracle Guided Image Synthesis with Relative Queries"
_ICLR.cc/2022/Workshop/DGM4HSD — ICLR 2022 DGM4HSD workshop Poster_

### Official Review · Reviewer_dAeD · 2022-03-23
**A novel method for guiding the image synthesis of VAEs using relative preferences from an oracle, backed up with convincing design choices and experiments**

**Rating:** 8
**Confidence:** 4

**Review:**

This is a very strong workshop paper that introduces a novel (to my knowledge) method for guiding VAE generation by: 1) using an additional triplet loss for pre-training the VAE; 2) using a Bayesian Triplet Loss objective for localizing the relative preferences of the oracle in an iterative fashion, after pre-training.

While reading the paper, two questions appeared, which were addressed by the authors later in the paper. Nevertheless, I post my questions below.

1. What happens when you let the reconstructive space to be non-zero-dimensional? How can you disentangle the relative and reconstructive spaces then? I am curious what the experiments will show, and whether modifications of your method would be necessary. I also wonder if you allow a non-zero-dimensional reconstructive space, whether you'd be able to decrease the reconstruction error in Table 1.

2. Why is the traditional method (in blue) much worse than the unsupervised method (in green) in panels (b-d) of Figure 3?

Important comments:

1. Experimental details are missing. I encourage the authors to share these details, so that people can reproduce the work. For example, how many queries did you use during localization?

Minor comments:

1. Isn't there a typo in Equation 2? Shouldn't you replace the $<$ sign with $-$?
2. $\phi$ should be a subscript of $q$ in Equation 3.
3. I don't think you formally define $\tau$ in Equation 4. Also, what is $m$?

---

### Decision · Program_Chairs · 2022-03-28

Accept (Poster)